# Evolution of the data and methods in real-world COVID-19 vaccine effectiveness studies on mortality: a scoping review protocol

Paulina Stehlik ,[1,2] Caroline Dowsett ,[1] Ximena Camacho,[3] Michael O Falster ,[3] Renly Lim ,[4] Sharifa Nasreen ,[5] Nicole L Pratt,[6] Sallie-Anne Pearson,[3] David Henry  [2,3]

For numbered affiliations see end of article.

**Correspondence to**
Paulina Stehlik;
p.stehlik@griffith.edu.au

## ABSTRACT

**Background** Early evidence on COVID-19 vaccine efficacy came from randomised trials. Many important questions subsequently about vaccine effectiveness (VE) have been addressed using real-world studies (RWS) and have informed most vaccination policies globally. As the questions about VE have evolved during the pandemic so have data, study design, and analytical choices. This scoping review aims to characterise this evolution and provide insights for future pandemic planning—specifically, what kinds of questions are asked at different stages of a pandemic, and what data infrastructure and methods are used?

**Methods and analysis** We will identify relevant studies in the Johns Hopkins Bloomberg School of Public Health VIEW-hub database, which curates both published and preprint VE RWS identified from PubMed, Embase, Scopus, Web of Science, the WHO COVID Database, MMWR, Eurosurveillance, medRxiv, bioRxiv, SSRN, Europe PMC, Research Square, Knowledge Hub, and Google. We will include RWS of COVID-19 VE that reported COVID-19-specific or all-cause mortality (coded as 'death' in the 'effectiveness studies' data set).

Information on study characteristics; study context; data sources; design and analytic methods that address confounding will be extracted by single reviewer and checked for accuracy and discussed in a small group setting by methodological and analytic experts. A timeline mapping approach will be used to capture the evolution of this body of literature.

By describing the evolution of RWS of VE through the COVID-19 pandemic, we will help identify options for VE studies and inform policy makers on the minimal data and analytic infrastructure needed to support rapid RWS of VE in future pandemics and of healthcare strategies more broadly.

**Ethics and dissemination** As data is in the public domain, ethical approval is not required. Findings of this study will be disseminated through peer-reviewed publications, conference presentations, and working-papers to policy makers.

**Registration** https://doi.org/10.17605/OSF.IO/ZHDKR

## STRENGTHS AND LIMITATIONS OF THIS STUDY

⇒ We will use a comprehensive curated database (Johns Hopkins Bloomberg School of Public Health VIEW-hub) that compiles relevant studies on a weekly basis from multiple databases, preprint servers, and the grey literature.
⇒ While use of a curated database may lead to some studies being missed, this is unlikely to change the overall findings of this scoping review.
⇒ All extraction will be conducted by a single author to ensure consistency in extraction and checked by a second author to ensure accuracy.
⇒ Weekly group discussions about the individual studies and coding of data will strengthen data integrity.
⇒ End users have been involved in the design of this study and will continue to be consulted throughout its conduct.

## INTRODUCTION

The COVID-19 pandemic has been unprecedented in terms of its direct health impacts and disruption of many aspects of modern society. It has also been remarkable in the speed with which scientists and industry collaborated in the production and testing of a range of vaccines.

It became apparent quickly that the COVID-19 vaccines did not stimulate sterilising immunity but provided protection against severe illness and death, most importantly in those with underlying risk factors.[1 2] The randomised trials that formed the evidence base for the initial deployment of vaccines included few subjects who were elderly, very young, pregnant, had immunodeficiency or severe comorbidity states.[3] Although quite large, the randomised trials documented few deaths and could not provide precise estimates of the effectiveness

of the vaccines in reducing COVID-related and all-cause mortality.

The subsequent evaluation of vaccine effectiveness (VE) using controlled observational studies has been complicated by changes in the infectiousness and virulence of the SARS-CoV-2 virus, and rising background levels of vaccine-induced or naturally acquired immunity. Case fatality rates have fallen substantially, particularly in highly vaccinated countries.[4] Deaths are now concentrated in a group of older patients, those with obesity and those who have serious comorbidities or are immunocompromised.[5] This rapidly changing landscape created a need for continuous 'real-world' studies (RWS) of VE in susceptible groups, against emerging viral variants and after repeated vaccine doses.[6] These studies use data collected outside of clinical trial settings to define exposures, endpoints, and relevant covariates. This is achieved by analysing data from electronic medical records, administrative records, death registries, and registries established specifically to record infection status and vaccine receipt.[6]

Most VE studies of COVID-19 vaccines have employed large population-scale linked routinely collected data sets. However, countries have varied in the timeliness of their response to this major challenge. In some countries, for instance Israel and UK, collaborations between researchers, health service providers, and government agencies enabled rapid analyses of large data sets using sophisticated techniques to adjust for confounding and other sources of bias. In contrast, other countries, for instance Australia and Aotearoa/New Zealand, were slow to conduct effectiveness studies, in part because of low infection rates early in the pandemic, and in Australia because of difficulties in accessing the necessary linked data sets.[7 8]

Systematic reviews of VE studies have concentrated, appropriately, on the vaccines' ability to prevent serious illness and death.[9–12] They have been consistent in confirming that multiple doses of the available vaccines have been associated with large reductions in mortality, with quite rapid waning (over months) in protection, mandating a need for repeated booster doses. As the impacts of vaccines on infection and transmission have been limited and transient,[13] it diminishes the value of infection as the principal study endpoint. The decline in PCR testing and registration of antigen test results have reduced the value of test results as the basis for test negative designs.[14 15] The nature of COVID-19-related hospitalisations has changed during the pandemic with an increase in incidental findings of infection through routine testing of patients admitted for other reasons.[15] On the other hand, there has been an increasing focus on excess all-cause mortality as a measure of the success of countries in controlling the spread of the virus and mitigating its negative impacts on healthcare systems.[16 17]

The COVID-19 pandemic has been a historic event that we must learn from. The rapid deployment of vaccines, followed by studies of their effectiveness, represents the largest and most important healthcare intervention in recent history and one that was evaluated largely using non-randomised studies. The sense of pandemic urgency led to rapid development of strategies to establish data sets, designs, and analytic approaches. This evolution of study questions, data designs, and methods through the course of the pandemic provides a unique learning opportunity for policy makers and researchers alike.

We plan to conduct a scoping review of the evidence base on real-world COVID-19 VE, focusing on studies that report on death as an outcome, to document this evolution. Specifically we will explore: how policy-relevant questions changed over the course of the pandemic, and how these affected the choices of data sources, designs, and analytical methods. By analysing these, we hope to provide information that is useful to the following stakeholders:

1. Policy makers and health system managers: by indicating what data sets will have to be created de novo and the need for linkage to existing routinely collected data in responding to future pandemics.
2. Clinicians and laboratory scientists: by identifying the disease manifestations and clinical and demographic vulnerability factors that will be required to inform the designs and analyses needed to evaluate the effectiveness of vaccines and other interventions, how these may change over the course of a future pandemic, and how the clinical community can advocate for the appropriate data elements to be linked and made available to researchers.
3. Data scientists and methodologists: to provide guidance as to study designs, analytical and adjustment techniques that are most often used in providing rapid estimates of VE early in a future pandemic; to advocate for the data elements required to deal with confounding to be collected and available in a linked analysable form.
4. Vaccine manufacturers: to understand better the postlicensing requirements for vaccines and pharmaceutical products under pandemic conditions and contribute appropriately to the necessary evaluations.
5. The pharmacoepidemiology community generally: the rapid evaluation of VE during the COVID-19 pandemic provides lessons for the timely investigation of a range of pharmaceutical treatments for emerging health threats.

## METHODS

We will conduct a scoping review, following the methods published by the Joanna Briggs Institute[18] and report the results according to the Preferred Reporting Items for Systematic Reviews and Meta-analyses statement for scoping review (PRISMA-ScR).[19] This scoping review is registered with the Open Science Framework (OSF; https://doi.org/10.17605/OSF.IO/ZHDKR). Data extraction has begun (25 September 2023, after protocol

registration), and will continue for approximately 12 months.

## Information sources and data selection

We will retrieve relevant studies from the VIEW-hub database,[20] maintained by Johns Hopkins Bloomberg School of Public Health. This database includes a wide range of study types including vaccine efficacy trials, VE studies, impact studies, and safety studies. As our principal aim is to describe the evolution of observational VE studies using real-world data, we used the VIEW-hub 'effectiveness studies' data set.

The VIEW-hub search strategy and inclusion criteria for this data set have been described in detail elsewhere (see online supplemental file).[21] Briefly, the 'effectiveness studies' data set includes both published and preprint studies of VE identified from PubMed, Embase, Scopus, Web of Science, the WHO COVID Database, MMWR, Eurosurveillance, medRxiv, bioRxiv, SSRN, Europe PMC, Research Square, and Knowledge Hub, as well as Google alerts for COVID-19 VE studies. Studies are screened weekly by the same two epidemiologists at Johns Hopkins Bloomberg School of Public Health, and the following data elements are extracted for studies included in the data set: study author, title, date published, link to paper, country of origin, vaccine studied, variant studied, population, study start and end date, and outcomes of interest. Studies in the data set can be filtered by the vaccine, variant, outcomes, study population, and region variables through drop-down menus.

Studies must also meet minimum criteria for causal inference studies using real-world data. The studies must include both vaccinated and unvaccinated (or other control) subjects, drawn from a comparable time period, capturing the relevant endpoints in both groups, having a secure record of vaccination (not relying on recall) and be free of obvious major methodological flaws. The latter judgement was not based on a strict risk of bias assessment.

To identify studies in the VIEW-hub's 'effectiveness studies' data set that examine mortality (either all-cause or cause-specific), we will use the drop-down menu feature to select study outcomes coded as 'death'. No additional eligibility criteria will be applied.

At the time of writing this protocol (1 August 2023), the VIEW-hub database lists 495 observational studies of VE from 50 countries, and 92 (~19%) list 'death' as an endpoint.

## Data extraction

We will extract data on:

1. *Study characteristics*: country, study design, publication status, protocol available, funding sources (including whether the study was funded by an independent source or manufacturer), study ethics approval (or waiver), consent requirements (or waiver).
2. *Study context*: reported vaccine policies in place, reported dominant viral variant at time of study.
3. *PICO-T*: inclusion and exclusion criteria, exposure (ie, vaccine(s)) and definition of exposure, control group, outcome definitions, outcomes collection period duration of follow-up and number of deaths.
4. *Data sources and additional variables*: the types of data sources used (eg, survey, electronic medical records, and administrative data), which were linked at an individual level and which were not, baseline confounders collected, and for adjusted outcomes which variables they were adjusted for.
5. *Analytical strategies to minimise bias*: methods for minimising baseline confounding (eg, propensity score analysis, instrumental variable analysis, covariate adjustment, self-controlled designs, etc) and further details of how the methods were implemented as appropriate, such as how the propensity score was implemented (matching, stratification, or inverse probability of treatment weights) and which variables were included in the propensity score model. Additionally, we will extract details on whether a sensitivity analyses was conducted, subgroups analysed, methods used for dealing with missing data, and methods used for dealing with time varying environmental risk.

We anticipate that there will be a few data points where it will be difficult to provide an exhaustive list of potential categories for some of the variables of interest a priori. We will therefore take an inductive approach to categorising variables such as 'data sources', 'inclusion criteria', and 'adjustment techniques' by entering them in free text and then developing categories through group discussion.

The lead author (PS) will develop a purpose-built data-extraction form in SharePoint Lists and a blank copy of the form and data dictionary will be provided on our OSF site. PS will also develop a validation set using a random sample of seven papers and verified by experts in pharmacoepidemiology (DH) and analysis (XC). A single author (CD) will independently extract data on the validation set until 80% agreement is achieved, at which stage they will continue with data extraction. A second reviewer (PS) will check the accuracy of all data extractions, and a core team (DH, CD, PS, and XC) will meet regularly to discuss each study, ensure it meets the inclusion criteria, and the main messages that it provides. The broader study team will meet less frequently to address issues arising and ensure data are categorised in a meaningful way that helps to inform decision making.

All data will be made publicly available via our study's OSF page (https://osf.io/m4cbf/).

## Assessment of risk of bias

We aim to describe the evolution of the literature and will therefore not conduct a formal assessment of the risk of bias in the included studies. However, all included studies in the VIEW-hub database must meet a minimal set of quality criteria, and while this does not mean that they are free of bias, the process aims to ensure a baseline level of quality.

### Data synthesis

To describe the evolution of RWS of COVID-19 VE over the course of the pandemic, we will use descriptive statistics to quantify study characteristics—including evolution of study designs (eg, test-negative designs, cohorts, and regression discontinuity), research questions asked (eg, comparisons of two doses vs boosters, effectiveness, and waning effect), data sources (eg, regularly collected population data and registry data), analytic approaches (eg, by design or form of adjustment), populations included, countries studied, outcome definitions, and event rates.

We will provide a temporal sequence of these characteristics overall, and where there are sufficient data within countries, present them visually (eg, as annotated stacked area graphs) to establish a template that enables anticipation of study questions and therefore supports planning for data availability in future pandemics.

We plan to develop interactive visuals as outputs so that stakeholders can interrogate the data further. All data manipulation, analysis, and visualisation will occur using Python and R and we will share all code via OSF.

### Review team and consultation

Our review team and reference group consist of content experts in review methodology, vaccine and drug effectiveness studies, biostatistics, and data science. Several have been involved directly in the conduct of VE studies during the COVID-19 pandemic and have a good working knowledge of the relevant literature. Most of the team members are actively involved in the National Health and Medical Research Council (NHMRC)-funded Centre for Research Excellence in Medicines Intelligence, which aims to accelerate real-world evidence development to inform medicines policy decision making.[22] Our reference group also comprises end users in infectious diseases and pandemic management, vaccine epidemiology, and medicines and vaccine policy.

All authors and advisory group members have provided comment on this protocol, and the appropriateness of the research questions and data elements. The advisory group will be consulted on how best to present the data so that it is usable and helps with decision making in each member's respective area.

In addition, we anticipate that the data we collect can be used for future review automation work and improve the efficiency of research. Our advisory group also includes an expert in review methodology and automation who will provide advice on future-proofing our dataset.

### Ethics and dissemination

As this scoping review will only include data in the public domain, ethics review is not required.

Findings of this review will be relevant to several stakeholders, including those involved in pandemic response, data infrastructure, and health technology evaluation. As such, we will disseminate our findings in five ways: (1) working papers for policy makers in Australia; (2) open access publication of findings in peer-reviewed journals; (3) presentation of findings at local and international infectious disease, vaccine, health systems, and health management conferences; (4) online interactive visual to allow interrogation of the extracted data; and (5) open access to our data, code, and preprints via OSF.

**Author affiliations**
¹School of Pharmacy and Medical Sciences, Griffith University Faculty of Health, Gold Coast, Queensland, Australia
²Institute for Evidence-Based Healthcare, Bond University Faculty of Health Sciences and Medicine, Gold Coast, Queensland, Australia
³School of Population Health, University of New South Wales Medicine & Health, Sydney, New South Wales, Australia
⁴Quality Use of Medicines and Pharmacy Research Centre, University of South Australia Division of Health Sciences, Adelaide, South Australia, Australia
⁵SUNY Downstate Health Sciences, University School of Public Health, New York, New York, USA
⁶Quality Use of Medicines and Pharmacy Research Centre, University of South Australia Clinical & Health Sciences Academic Unit, Adelaide, South Australia, Australia

**Acknowledgements** We would like to acknowledge the advisory group and Oyungerel Byambasuren to their feedback and comments on draft versions of this document, particularly on the methodology and which variables to collect to provide meaningful information to decision makers.

**Contributors** PS and DH conceptualised the project, acquired the funding, and are acting as project supervisors. PS, CD, XC, MOF, RL, SN, NLP, S-AP, and DH contributed to the methodology. PS developed the resources and database and will oversee database and project management. PS, DH, and XC piloted the database and extraction tool and developed the validation set. CD is conducting the data extraction which will be checked by PS. PS, XC, and MOF developed the data synthesis plan. PS and DH wrote the original draft of this manuscript, and PS, CD, XC, MOF, RL, SN, NLP, S-AP, and DH all edited and reviewed the draft and final revisions.

**Funding** Medicines Intelligence Centre for Research Excellence (MI-CRE) 2022 Project Incubator Grant; The MI-CRE is supported by the National Health and Medical Research Council's Centres of Research Excellence (CRE) scheme (ID 1196900). RL is supported by a National Health and Medical Research Council (NHMRC) Early Career Fellowship (APP1156368). MOF is supported by a Future Leader Fellowship from the National Heart Foundation of Australia (ID: 105609). XC is supported by a NHMRC Postgraduate Scholarship (ID: 2005259).

**Competing interests** None declared.

**Patient and public involvement** Patients and/or the public were not involved in the design, or conduct, or reporting, or dissemination plans of this research.

**Patient consent for publication** Not applicable.

**Provenance and peer review** Not commissioned; externally peer reviewed.

**ORCID iDs**
Paulina Stehlik http://orcid.org/0000-0002-5397-228X
Caroline Dowsett http://orcid.org/0000-0001-7734-9436

Michael O Falster http://orcid.org/0000-0001-6444-7272
Renly Lim http://orcid.org/0000-0003-4135-2523
Sharifa Nasreen http://orcid.org/0000-0002-9793-113X
David Henry http://orcid.org/0000-0003-2934-2242

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
