## [Reviewer comments · BMJ Open]

ARTICLE DETAILS

TITLE (PROVISIONAL)	EVOLution of the data and methods in real world COVID-19 Vaccine Effectiveness studies on mortality: A Scoping Review Protocol
AUTHORS	Stehlik, Paulina; Dowsett, Caroline; Camacho, Ximena; Falster, Michael; Lim, Renly; Nasreen, Sharifa; Pratt, Nicole; Pearson, Sallie-Anne; Henry, David

VERSION 1 – REVIEW

REVIEWER	Skoetz, Nicole University Hospital of Cologne , Cochrane Haematological Malignancies Group; Department I of Internal Medicine
REVIEW RETURNED	29-Aug-2023

GENERAL COMMENTS	Thank you for this very important protocol My main concern is related to the search in one database only and inclusion criteria, It remains unclear who evaluates pre-specified quality criteria to be included in this database, also coding of outcomes (deaths). The criteria listed do not seem to be related to quality only, but to content of the paper and published results. Will studies be excluded if no deaths occurred? No deaths could be an important outcome of a vaccine study. Moreover, it remains unclear what is meant by real world data. Will RCTs not be included? I agree to include observational studies, in addition, the intervention arm of RCTs could also add valuable information. According to the definition of a scoping review I suggest to include a broad search strategy and inclusion criteria, not to miss relevant studies and to provide a huge overview of relevant literature. minor comment: related to data extraction: an agreement of 80% is not very high, I suggest double data extraction or a higher threshold
---

REVIEWER	Ray, Arindam Bill and Melinda Gates Foundation India
REVIEW RETURNED	14-Sep-2023

GENERAL COMMENTS	VE needs to be defined clearly and consistently, while studies are considered for inclusion. Keeping in mind the evolving landscape of covid19 vaccine studies, the studies on durability of protection, immune imprinting, breakthrough infections and interaction between episodes of infection followed, preceded or punctuated
--

	by variable number of vaccine doses, need to be considered for inclusion.
--	---

VERSION 1 – AUTHOR RESPONSE

Reviewer: 1

Dr. Nicole Skoetz, University Hospital of Cologne

Thank you for this very important protocol.

My main concern is related to the search in one database only and inclusion criteria, It remains unclear who evaluates pre-specified quality criteria to be included in this database, also coding of outcomes (deaths).

We thank Dr Skoetz for her careful reading of the protocol and insightful comments. VIEW-hub is a curated database of COVID19 vaccine effectiveness studies is a result of systematic searches of several databases, preprint servers and sources of potential grey literature. For clarity we have added the following to the “information and data sources” section:

“The VIEW-hub was established in 2016 as a go-to resource for researchers, decision makers and funders, policy makers and advocates for reliable vaccine information. Since early 2021 it has conducting systematic searches of studies of COVID-19 vaccine effectiveness on a weekly basis, and has been used by researchers, regulators and policy makers to evaluate COVID-19 vaccine effectiveness previously...The VIEW-hub search strategy and inclusion criteria have been described in detail by VIEW-hub curators and the database is updated weekly. Broadly speaking the database includes both published and pre-print studies of vaccine effectiveness identified from PubMed, Embase, Scopus, Web of Science, the WHO COVID Database, MMWR, Eurosurveillance, and medRxiv, bioRxiv, SSRN, Europe PMC, Research Square, and Knowledge Hub, as well as Google alerts for COVID-19 vaccine effectiveness studies”.

In our responses to the Editors, we have provided our rationale for the use of VIEW-hub’s database of COVID19 vaccine effectiveness studies.

We have also added text in the main body of the manuscript describing the reported methods for selection of studies and any assessments of study quality in the View-hub database:

“Studies are screened weekly by the same two epidemiologists, who also extract some data about included studies. These include study author, title, date published, link to paper, country of origin, vaccine studied, variant studied, population, study start and end date, and outcomes of interest.”

The criteria listed do not seem to be related to quality only, but to content of the paper and published results.

Thank you for this comment. As we have mentioned above this is a scoping review. We did not make a formal assessment of risk of bias (RoB) of candidate studies. In assembling the VIEW-Hub database minimum methodological criteria were applied, as noted below. Our only additional filter was to confine selection to those studies that included mortality as an endpoint. We have added the following to our manuscript: *“To be included in the VIEW-Hub database studies must include at least one vaccine effectiveness estimate and meet a minimal set of quality criteria (e.g. studies must have a contemporaneous control, COVID-19 must be confirmed through PCR or antigen test,*

vaccination status cannot be established via recall, and studies must have no significant bias that likely affects results).”

Will studies be excluded if no deaths occurred? No deaths could be an important outcome of a vaccine study.

We agree with this entirely and studies that report zero deaths will be included. Any study that aimed to include deaths as an endpoint will be included, irrespective of the numbers of deaths reported.

It is worth noting that we do not plan on meta-analysing the outcomes but rather will report on how they were defined and measured and what data and methods were used.

Moreover, it remains unclear what is meant by real world data. Will RCTs not be included? I agree to include observational studies, in addition, the intervention arm of RCTs could also add valuable information.

As we note in the background the randomised trials of vaccines were conducted early in the pandemic and demonstrated short term immune responses and protection against disease. It quickly became unethical to use randomised trials to address the many emerging questions about targeting of vaccines to vulnerable groups, waning protection and effectiveness against new viral variants. The great majority of evidence that has guided vaccination policies came from controlled observational studies that relied on analysis of linked routinely collected data. This is what we mean by ‘real world data’. As our aim is to map the evolution of these studies rather than that from randomised trials, randomised trials will not be included.

To clarify our definition of real-world data, we have added the following sentence in the background:

“This rapidly changing landscape created a need for continuous ‘real-world’ studies (RWS) of vaccine effectiveness in susceptible groups, against emerging viral variants and after repeated vaccine doses.(2) These studies use data collected outside of a clinical trials setting to define exposures, endpoints and relevant covariates. This is achieved by analysing data from electronic medical records, administrative records, death registries and registries established specifically to record infection status and vaccine receipt.(2)”

According to the definition of a scoping review I suggest to include a broad search strategy and inclusion criteria, not to miss relevant studies and to provide a huge overview of relevant literature.

We agree with this statement. As described earlier, we believe that the VIEW-hub database curators conduct a thorough and broad search of the literature directly relevant to our research question. This includes several databases and preprint servers as well as the grey literature which we describe in the methods section:

“Broadly speaking the database includes both published and pre-print studies of vaccine effectiveness identified from PubMed, Embase, Scopus, Web of Science, the WHO COVID Database, MMWR, Eurosurveillance, and medRxiv, bioRxiv, SSRN, Europe PMC, Research Square, and Knowledge Hub, as well as Google alerts for COVID-19 vaccine effectiveness studies.”

As we seek to describe the evolution of COVID-19 effectiveness studies conducted using real world data with a focus on describing the methods, data and analytic techniques used, we believe that the quality criteria for inclusion into the curated set is highly appropriate and relevant. Describing the evolution of poor-quality studies that did not further our understanding of vaccine effectiveness in any meaningful way would not meet the aims of our research question.

minor comment: related to data extraction: an agreement of 80% is not very high, I suggest double data extraction or a higher threshold

We thank you for your suggestion. We have used a level of 80% agreement with a sample of the studies is a threshold set by the AMSTAR group.(3). In addition and in line with recommendations with JBI Scoping Review methodology recommendations,(4) we will be holding regular (weekly) meetings to discuss the data extracted and to ensure its correctness and completeness, and all data entry will be checked by a second reviewer (PS). We have added the following to our manuscript under data extraction:

“A second reviewer (PS) will check the accuracy of all data extractions, and a core team (DH, CD, PS, XC) will meet regularly to discuss each study, whether it meets the inclusion criteria, and the main messages that it provides.”

We believe that this is sufficient to ensure robustness in the data.

Reviewer: 2

Dr. Arindam Ray, Bill and Melinda Gates Foundation India

VE needs to be defined clearly and consistently, while studies are considered for inclusion. Keeping in mind the evolving landscape of covid19 vaccine studies, the studies on durability of protection, immune imprinting, breakthrough infections and interaction between episodes of infection followed, preceded or punctuated by variable number of vaccine doses, need to be considered for inclusion.

Thank you for this comment. We agree that these are important considerations. One of the aims of this scoping review is to chart the evolution of these challenges to vaccine effectiveness during the pandemic and how they were addressed in terms of study designs, data, methods and analyses.

We are not estimating VE in this study and will not define vaccine effectiveness measures ahead of the review. Rather we will document and analyse the definitions were used by investigators, bearing in mind that our focus at this stage is on fatal outcomes.

References

1. Feikin DR, Higdon MM, Abu-Raddad LJ, Andrews N, Araos R, Goldberg Y, et al. Duration of effectiveness of vaccines against SARS-CoV-2 infection and COVID-19 disease: results of a systematic review and meta-regression. *The Lancet*. 2022;399(10328):924-44.
2. Swift B, Jain L, White C, Chandrasekaran V, Bhandari A, Hughes DA, et al. Innovation at the Intersection of Clinical Trials and Real-World Data Science to Advance Patient Care. *Clinical and Translational Science*. 2018;11(5):450-60.

3. Shea BJ, Reeves BC, Wells G, Thuku M, Hamel C, Moran J, et al. AMSTAR 2: a critical appraisal tool for systematic reviews that include randomised or non-randomised studies of healthcare interventions, or both. *BMJ*. 2017;358:j4008.
4. Pollock D, Peters MDJ, Khalil H, McInerney P, Alexander L, Tricco AC, et al. Recommendations for the extraction, analysis, and presentation of results in scoping reviews. *JBI Evidence Synthesis*. 2023;21(3):520-32.

VERSION 2 – REVIEW

REVIEWER	Skoetz, Nicole University Hospital of Cologne , Cochrane Haematological Malignancies Group; Department I of Internal Medicine
REVIEW RETURNED	19-Nov-2023

GENERAL COMMENTS	Thank you for the revised version I do have some major comments, related to inclusion criteria As a scoping review should provide a broad overview of available literature, it does not seem to be comprehensive, if some studies are excluded based on the decision of two epidemiologists. You mention the inclusion criterion: studies must have a contemporaneous control. How do you define contemporaneous control in terms of vaccines during a pandemic? I guess it will be no vaccination at the beginning of the pandemic, but should be an effective vaccination nowadays? Please elaborate and list included control groups over time which will be included As bias assessment is very subjective: how do you define "no significant bias"? Are unblinded studies excluded? Or only if outcome assessment was not blinded? How will you handle attrition bias, selection bias? As nowadays bias is often assessed on outcome level: how do you consider this in your inclusion decision? Which bias tool used the two epidemiologists? Is their assessment available for interested readers, especially for excluded studies? Otherwise, the scoping review will not be very comprehensive
---

VERSION 2 – AUTHOR RESPONSE

We are grateful to your referee for their further reading of our paper and will attempt to address the outstanding issues.

1) As a scoping review should provide a broad overview of available literature, it does not seem to be comprehensive, if some studies are excluded based on the decision of two epidemiologists.

a. The full VIEW-hub database, developed by JHBSPH in collaboration with WHO, is very large, covering COVID and non-COVID vaccines and including information on vaccine characteristics, international coverage levels, vaccine impacts, vaccine efficacy, vaccine effectiveness, vaccine safety

and immune reactivity. We have no role in compiling this database and are accessing only the subset of studies concerned with COVID vaccine effectiveness, and then focusing on those studies with mortality as an endpoint. Because these studies are making causal inferences about the direct effect of vaccination, they must meet certain minimum criteria to be included in that section of the VIEW-hub database. This initial screen requires that studies include both vaccinated and unvaccinated (or other control) subjects and capture the relevant endpoints in both groups, have a secure record of vaccination (not relying on recall) and are free of obvious major methodological flaws. This is an eligibility screen, not a risk of bias assessment. Studies that have used this database in systematic reviews have found methodological flaws in the component studies, indicating that the bar for inclusion was (appropriately) set low. Our co-authors and members of our reference group have published over a dozen COVID vaccine effectiveness studies in peer-reviewed journals, so we are very familiar with this literature. At present the VIEW-hub database includes 528 vaccine effectiveness studies, conducted over 3 years, which is unprecedented for any healthcare intervention. As the reviewer states we are performing a scoping review, which is focused not on the results but designs, data sources and methods. Consequently, our findings are unlikely to be sensitive to omission of individual studies. Because of the sheer size of this literature, we did not have the resources to perform a primary literature search, and this would have been a wasteful exercise. We have clarified these issues in further edits to the protocol (Page 6, paragraphs 1-2)

2) You mention the inclusion criterion: studies must have a contemporaneous control. How do you define contemporaneous control in terms of vaccines during a pandemic? I guess it will be no vaccination at the beginning of the pandemic, but should be an effective vaccination nowadays? Please elaborate and list included control groups over time which will be included.

a. Because the VE studies were conducted during a pandemic the control group had to be selected during the pandemic to be at risk of the relevant study outcomes and also be eligible for vaccination. As the reviewer has correctly stated there are more detailed temporal considerations including important time varying factors such as environmental risk (transmission rates) and variant waves. However, these are not selection factors for inclusion in the database, nor in our review, but are examples of the types of information that will be collected during the scoping review.

b. The question of the nature of the control exposure is a related but different issue. As the reviewer has stated, this changed through the pandemic. During the later variant waves the control exposure was sometimes 2 doses while the intervention was three, four or more doses with inclusion of bivalent and more recently monovalent variant-modified vaccines. Some studies compared the effectiveness of different vaccines (active comparator studies). These are not inclusion/exclusion factors but, as above, represent the types of information we will extract during the review.

3) As bias assessment is very subjective: how do you define "no significant bias"? Are unblinded studies excluded? Or only if outcome assessment was not blinded? How will you handle attrition bias, selection bias? As nowadays bias is often assessed on outcome level: how do you consider this in your inclusion decision? Which bias tool used the two epidemiologists? Is their assessment available?

a. As we noted earlier vaccine effectiveness studies conducted during a pandemic have certain minimum requirements and these determined eligibility for inclusion in the VIEW-hub database but did not guarantee their overall methodological quality. As noted above full RoB assessment was subsequently carried out by some researchers performing meta-analysis and meta-regression analyses of the results (see Feikin DR et. al. Lancet 2022; 399: 924–44 as an example). We will not be performing RoB assessment as we are not planning to perform meta-analysis of the results of the VE studies – this has been carried out by other groups. In our scoping review we are focusing on the

data, design and analytic strategies and how these evolved during the pandemic. In terms of blinding – these are not experimental studies, so blinding of observers or participants is not possible. Most studies employ large routinely collected databases. As noted above we have edited the protocol to make this clearer.